# N-3 Long-Chain Polyunsaturated Fatty Acids, Eicosapentaenoic and Docosahexaenoic Acid, and the Role of Supplementation during Cancer Treatment: A Scoping Review of Current Clinical Evidence

**DOI:** 10.3390/cancers13061206

**Published:** 2021-03-10

**Authors:** Marnie Newell, Vera Mazurak, Lynne M. Postovit, Catherine J. Field

**Affiliations:** 1Department of Agricultural, Food and Nutritional Science, Faculty of Agricultural, Life and Environmental Sciences, University of Alberta, Edmonton, AB T6G 2E1, Canada; marnie.newell@ualberta.ca (M.N.); vmazurak@ualberta.ca (V.M.); 2Department of Oncology, Faculty of Medicine and Dentistry, University of Alberta, Edmonton, AB T6G 2R7, Canada; postovit@ualberta.ca; 3Department of Biomedical and Molecular Sciences, Queen’s University, Kingston, ON K7L 3N6, Canada

**Keywords:** docosahexaenoic acid (DHA), eicosapentaenoic acid (EPA), clinical, intervention, immune, outcomes

## Abstract

**Simple Summary:**

There has been extensive research into the beneficial anticancer effects of n-3 long-chain polyunsaturated fatty acids (LCPUFA), eicosapentaenoic acid (EPA) and docosahexaenoic acid (DHA) in preclinical models of cancer. However, clinical evidence is limited. The aim of this scoping review was to summarize the current clinical evidence of n-3 LCPUFA supplementation in cancer treatment and highlight areas where more clinical evidence is needed. We summarized the results of 57 clinical trials with an EPA/DHA intervention and determined that supplementation could improve a variety of outcomes important to the patient and the disease process, including immune system modulation, improved weight maintenance and increased disease-free or progression-free survival. There is, however, a need for larger, well-controlled, statistically powered randomized controlled trials to move n-3 supplementation to clinical practice.

**Abstract:**

This scoping review examines the evidence for n-3 long-chain polyunsaturated fatty acid [LCPUFA, eicosapentaenoic acid (EPA) and docosahexaenoic acid (DHA)] supplementation in clinical cancer therapy. A comprehensive literature search was performed to identify relevant clinical intervention studies conducted through August 2020. Fifty-seven unique cancer trials, assessing EPA and/or DHA supplementation pre- or post-treatment, concomitant with neoadjuvant chemotherapy, radiation or surgery, or in palliative therapy were included. Breast, head and neck, gastrointestinal, gastric, colorectal/rectal, esophageal, leukemia/lymphoma, lung, multiple myeloma and pancreatic cancers were investigated. Across the spectrum of cancers, the evidence suggests that supplementation increased or maintained body weight, increased progression-free and overall survival, improved overall quality of life, resulted in beneficial change in immune parameters and decreased serious adverse events. Taken together, the data support that EPA and/or DHA could be used to improve outcomes important to the patient and disease process. However, before incorporation into treatment can occur, there is a need for randomized clinical trials to determine the dose and type of n-3 LCPUFA intervention required, and expansion of outcomes assessed and improved reporting of outcomes.

## 1. Introduction

In 2020, an estimated 19.3 million cases of cancer were diagnosed worldwide. The most frequently diagnosed cancers across both sexes was breast (11.7%), followed by lung (11.4%), colorectal (10%) and prostate cancer (7.3%) [1]. Despite advances in diagnosis and treatment, cancer accounted for an estimated 10 million deaths globally in 2020 [1]. Cancer is the second leading cause of death in the United States [2] and the leading cause of death in Canada [3]. Improving current conventional therapies and treatment paradigms could result in improved patient outcomes and a reduction in deaths. As such, there has been extensive research into the efficacy of n-3 long-chain polyunsaturated fatty acids (LCPUFA), eicosapentaenoic acid (EPA) and docosahexaenoic acid (DHA) in preclinical models of cancer. The pleiotropic effects of n-3 LCPUFA on tumors include increasing apoptosis, inducing cell cycle arrest, decreasing cell growth, and halting proliferation in experimental models of cancer (reviewed in [4,5,6]).

While there is a growing body of strong preclinical evidence, evidence in patient populations is limited. The difficulty in translating laboratory findings to a clinical setting arises from tremendous heterogeneities that exist within tumors (intra-tumoral) and between patients which cannot readily be replicated in immortalized cancer cell models. Moreover, the role of the tumor microenvironment, including cells of the immune system is difficult to assess in immunocompromised animal models. Furthermore, side effects from cancer and the treatment for cancer have not been properly assessed in animal models. This includes pain, weight loss, quality of life (QOL) and peripheral neuropathy or other chemotherapy-specific adverse effects. The aim of this scoping review is to examine the current clinical evidence on n-3 LCPUFA supplementation in cancer treatment and highlight areas where more clinical evidence is needed.

## 2. Materials and Methods

### 2.1. Study Inclusion and Exclusion Criteria

Studies from peer-reviewed literature were included if they involved cancer patients where n-3 LCPUFAs (EPA and/or DHA) were provided as the intervention during cancer treatment. Review papers, abstracts, individual case reports and animal or cell line studies were excluded.

### 2.2. Data Sources, Search Strategy and Study Selection

A systematic literature search strategy was used to identify relevant articles. We searched electronic bibliographic databases including Ovid MEDLINE and EBSCOhost CINAHL from each database inception through August 2020. Searches were restricted to peer-reviewed studies. Search terms were searched as MeSH headings or keywords in title or abstract and were derived from (1) DHA/EPA/dietary fats (2) immune response/inflammation and (3) cancer/solid tumors/oncology treatment. Titles and abstracts of all citations were screened based on predefined eligibility and duplicates were removed. Full texts of eligible abstracts were exported to EndNote and underwent a second screen by two investigators and were verified for inclusion eligibility.

### 2.3. Data Extraction and Synthesis

Data extraction was completed by MN and reviewed for accuracy by the primary investigator (CJF). Data extraction included manuscript information (authors, title, year of publication), study characteristics (primary objective, study design, cancer localization and stage), patient characteristics [number enrolled/completed, mean age, gender, ethnicity, body weight/body mass index (BMI)] intervention characteristics (chemotherapy/radiation/surgical treatment, treatment length, n-3 LCPUFA type: capsules, oral supplementation, enteral/parenteral nutrition, intervention length relative to treatment length, frequency of administration, control characteristics, assessment of compliance), and outcome characteristics (quantitative and qualitative assessments). 

## 3. Results

### 3.1. Study Selection

The initial database searches yielded 729 distinct citations from Ovid MEDLINE and 237 from EBSCOhost CINAHL. Twenty papers were found to be duplicates and removed. After evaluation of the titles and abstracts, an additional 707 articles were excluded for failing to meet the inclusion and exclusion criteria. After screening the abstracts based on the eligibility criteria, 147 papers were exported from Ovid MEDLINE and 92 from EBSCOhost CINAHL, resulting in a total of 239 articles. An additional 182 articles were excluded as they failed to meet the criteria resulting in a final number of 57 studies included in the current review (Figure 1). 

### 3.2. Overview of Studies 

A spectrum of cancers including breast [7,8,9,10,11,12,13,14,15,16], head and neck [15,17,18,19], gastrointestinal [14,20,21,22,23,24,25,26,27], gastric [21,28,29,30,31], colorectal/rectal [21,32,33,34,35,36,37,38,39,40,41,42,43,44,45], esophageal [15,19,21,46,47], leukemia/lymphoma [48], lung [14,15,21,40,49,50,51,52,53,54,55], multiple myeloma [56] and pancreatic [14,21,55,57,58,59,60,61,62,63] have been investigated for efficacy of n-3 supplementation pre- or post-treatment [12,13], concomitant with neoadjuvant chemotherapy or radiation [7,8,9,10,11,12,15,16,18,19,20,21,26,27,30,31,32,33,34,37,45,46,48,49,50,51,53,54,56,62,63,64], in conjunction with surgery [17,18,22,23,24,25,28,35,38,42,44,46] or during palliative therapy [14,36,39,40,41,47,52,57,58,59,60,61,64,65]. Furthermore, DHA and EPA treatments in a clinical setting have been delivered through multiple modalities including capsules: Table 1 and Table 2 [7,8,9,10,11,12,13,14,16,20,26,27,30,32,33,34,35,36,45,48,49,50,51,52,56,57,58,63], oral: Table 3 [15,17,18,21,31,37,38,39,40,41,53,54,55,59,60,61,64] and enteral/parenteral supplementation: Table 4 [19,22,23,24,25,28,29,42,43,44,46,47,62,65]. Much of the current body of research has occurred in surgical or palliative patients receiving oral or enteral/parenteral nutrition that is routinely employed in cancer patients to provide nutritional support, especially in instances of weight loss and cachexia or during surgical interventions. 

### 3.3. N-3 Type, Amount Prescribed and Intervention Length

Of the 57 independent studies assessed, two trials reported supplementation with DHA alone (in triglyceride form from an algal source [7] or titrated from fish oil [56], two studies employed EPA alone (as free fatty acids [36,57] and 48 studies used a combination of EPA and DHA (derived from fish oil). An additional five studies reported only supplementing with EPA, yet based on the enteral supplement reported, these interventions likely included DHA but did not explicitly state it in the manuscript. Across the spectrum of studies, supplementation varied greatly in concentrations and intervention length. When the supplement was provided in an oral capsule form (Table 1 and Table 2), supplementation ranged from 300 milligrams (mg) to 5 grams (g) of n-3 fatty acids with 77% of the studies providing 2 g or less per day. Of the total supplementation, DHA concentrations ranged from 0 to 2 g, with 86% of the studies providing 1 g or less per day. When capsules were provided concomitant with chemotherapy the intervention length was 6–24 weeks and equivalent to the duration of chemotherapy [7,8,9,10,11,16,20,26,27,30,31,32,33,34,45,48,49,50,51,56,63]. In the absence of chemotherapy, supplementation was provided for 1–6 months [12,13,14,35,36,52,57]. Oral supplementation interventions were one week to 6 months in length, with over half of the studies ranging from 4 to 6 weeks in duration. Ten of the studies provided similar doses of EPA and DHA (2.2 and 0.9 g per day, respectively) with a range of 2–3.6 g of total n-3 per day (1.1–2.4 g EPA ± 0.9–1.2 g DHA) and in one study, 4.9 g of EPA + 3.2 g of DHA per day (Table 3). The amount of n-3 fatty acids provided by enteral/parenteral nutrition was variably reported in the assessed studies; in some instances, 0.2 g n-3 per kilogram (kg) of body weight and in others from 2.25 to 3.3 g per day (Table 4). In these studies, the interventions were acute, generally limited to before and after surgery and were approximately 5–9 days in duration.

### 3.4. Outcomes

Reported outcomes varied across the studies but are grouped below into the following categories: (1) weight gain or maintenance, (2) serious adverse events including neuropathy and length of hospital stay, (3) immunological measures, (4) quality of life, (5) overall survival or progression-free survival and (6) additional parameters. Figure 2 provides an overall summary of the findings. 

#### 3.4.1. Weight

Weight loss is a common side effect of both cancer and cancer therapies and has been reported on in many studies investigating the efficacy of n-3 supplementation particularly in advanced or palliative cancers and ones with defined pre-study weight loss. In general, oral liquid supplementation (which also included protein) was provided in instances where there was substantial weight loss prior to study entry and some evidence of malnutrition or cachexia. Compared to baseline status, n-3 supplemented groups achieved weight gain in colorectal, gastric and pancreatic cancer studies [31,37,55,59,60,61,63] or weight maintenance in one head and neck cancer study [18]. Two studies, one in advanced gastrointestinal cancer patients and one in stage III non-small-cell lung cancer patients observed improved weight maintenance in n-3 supplemented groups compared to control patients who were provided an isocaloric supplement that was not fortified with n-3 fatty acids [39,53]. 

Capsule supplementation has been employed in three studies with palliative patients where weight maintenance was a reported study metric. In advanced lung cancer patients with systemic immune metabolic syndrome (SIMS, defined by presence of cachexia, anorexia, ECOG (Eastern Cooperative Oncology Group) status > 2 and high CRP), the combination of 600 mg EPA + DHA and celecoxib (non-steroidal anti-inflammatory drug) increased body weight by approximately 1.2 kg compared to the n-3 capsules alone over a 6-week intervention [52]. Pancreatic cancer patients, who at study entry had lost approximately 13% body weight, stabilized their weight, and began to increase it by week 4 with EPA supplementation and the authors suggest that EPA could be a safe, effective anticachectic agent that could result in weight gain [57]. Finally, in a low-dose EPA + DHA (300 mg total per day) study of palliative pancreatic cancer patients, where two different sources were provided (fish oil or marine phospholipid capsules), similar effects were observed between the 2 groups suggesting the efficacy of n-3 fatty acids in weight stabilization [58].

Four studies reported weight maintenance or gain when n-3 fatty acids capsules were provided with neoadjuvant chemotherapy. In advanced lung cancer patients supplemented with 3.4 g of EPA + DHA concomitant with gemcitabine and cisplatin chemotherapy, body weight increased by 3.4 kg over the 66 day intervention [50]. Gastrointestinal patients supplemented with 700 mg of EPA + DHA for 8 weeks with 5-fluorouracil and leucovorin chemotherapy gained an average of 1.7 kg over 8 weeks and although not significant to their baseline status, was significantly different from control patients who lost 2.5 kg average during this time frame [20]. Haidari et al. reported that colorectal cancer patients supplemented with 608 mg of EPA + DHA resulted in weight maintenance over 8 weeks compared to control patients [27]. In colorectal cancer patients treated with xeloda, oxaliplatin, 5-fluorouracil and/or leucovorin therapy, it was observed that 600 mg of EPA + DHA per day potentially prevented weight loss over 9 weeks when compared to standard of care control [32]. 

#### 3.4.2. Serious Adverse Events

No studies reported serious adverse events attributable to n-3 supplementation, although in two instances with surgical patients for gastric and gastrointestinal cancers, there was improvement in overall postoperative recovery [29] and length of hospital stay [25], respectively, with parenteral n-3 nutrition. Lung cancer patients supplemented with up to 2.7 g of EPA + DHA were observed to have a better chemotherapy response rate during neoadjuvant therapy [49]. Common side effects of cytotoxic therapies used in breast cancer treatment include febrile neutropenia and neuropathy [66] and two studies observed a beneficial effect of supplementation on their side effects during neoadjuvant chemotherapy. Metastatic patients supplemented with 1.8 g of DHA per day had decreased neutropenia, anaemia and thrombopenia during cytotoxic chemotherapy [7]. Additionally, when women were supplemented with 1.2 g of EPA + DHA, Ghoreishi et al. observed a 70% decreased risk of peripheral neuropathy incidence over 4 cycles of paclitaxel treatment [9]. These beneficial effects were also observed in palliative esophageal cancer patients, where EPA + DHA supplementation resulted in decreased nausea, thromboembolism, leucopenia, and neutropenia [47].

#### 3.4.3. Immunological Outcomes

Enteral or parenteral nutrition is commonly employed for surgical interventions, resulting in elevated inflammation. All studies assessed in this review reported immunological modulations resulting from n-3 fatty acid-enriched enteral/parenteral nutrition. This included improved immune cell response [28,42,43,46]; decreased interleukin (IL)-8 [46,65], IL-10 [46] and IL-6 (at time points 8–21 days post-surgery compared to control) [23,24,28,29,42,43,44,46]; modulation of functional capacity and gene expression of immune markers [19]; increased T-lymphocytes, T helper and natural killer (NK) cells [24]; modulated cytokine production [23,65]; decreased prostaglandin E2 (PGE2) [22] and reduced incidence of systemic inflammatory response syndrome [25,44]. C-reactive protein (CRP), a marker of inflammation that is often used as an indicator of poor prognosis, was routinely elevated post-surgery but found to decrease during n-3 supplementation in the days following [24,46]. 

Chemotherapy alters the immune response and inflammatory status [67] yet evidence of beneficial immunomodulation with n-3 supplementation in non-surgical settings is limited. CRP was the most frequently assessed marker of inflammation, most often in advanced or palliative cancer patients, where it was observed that capsule/oral n-3 supplementation decreased CRP in head and neck [18], lung [40,52,54], gastrointestinal [27,40,64], or pancreatic [64] cancers or maintained CRP levels during n-3 supplementation in breast [12,13], lung [50] or pancreatic cancer [57] compared to either baseline levels or increased CRP in non-supplemented controls (Appendix A
Appendix A). Additionally, the CRP/albumin ratio, believed to be a predictor of overall survival in many cancers [68], was decreased during n-3 supplementation [32,33,48]. Other markers reported to be beneficially decreased during n-3 supplementation include IL-6 [27,31,45,50], PGE2 [15,50], tumor necrosis factor alpha (TNFα) [18,27,47,54] and interferon gamma (IFNγ) [13,18,21]. Purasiri et al. [35], assessed both localized and advanced colorectal cancer patients. In the patients with localized cancer, supplementation with 1 g EPA + 160 mg DHA per day, short term until surgery, had no observed changes in immune parameters. However, in the advanced patients, where the amount of EPA + DHA increased to 2.1 g EPA and 320 mg DHA daily for months 2 to 6, a decrease in IL-1β, IL-4, IL-6, TNFα, and IFNγ was observed. Interestingly, no changes in cytokine concentrations occurred in the first 2 months, and the authors suggested that long-term supplementation results in a significant decrease in circulating cytokines [35]. Additionally, neutrophil function during chemotherapy improved [20], cluster of differentiation (CD)4/CD8 ratio increased [14] and percent of CD4 + and CD8 + was maintained compared to elevation in controls [12]. 

#### 3.4.4. Quality of Life

Changes in life quality have been frequently reported in clinical trial outcomes and often studies have employed the validated questionnaire from the European Organisation for Research and Treatment of Cancer-Quality of Life Questionnaire-C30 [69]. In capsule or oral n-3 supplementation studies, improved perceived quality of life was reported in five studies [26,37,45,54,55] and improved appetite reported twice—in colorectal and non-small-cell lung cancer studies [37,54]. Musculoskeletal pain is a well-documented side effect of aromatase inhibitors [70]. In women with previous chemotherapy or radiation and currently on aromatase inhibitors for breast cancer therapy, 1.4 g of n-3 per day resulted in a 21.5% decrease on a pain scale after 30 days [13]. However, a second study in breast cancer patients on aromatase inhibitors found that supplementation with 3.3 g of EPA + DHA for 24 weeks only decreased pain significantly in obese patients [10,11]. 

#### 3.4.5. Survival

Overall survival, progression-free survival and disease-free survival are key metrics reported in clinical trials. In the current review of the literature, studies investigating the benefits of n-3 supplementation via enteral/parenteral nutrition did not report on these metrics, likely due to the acute time frame of the interventions. However, a clinical benefit was observed in other studies. Stage III/IV lung cancer patients who received 1.1 g of EPA in a protein enriched oral supplement were reported to have a trend towards progression-free survival (*p* < 0.07) [54]. In metastatic patients with a variety of cancers, high doses of n-3 (5 g EPA + DHA combo) increased survival (*p* < 0.02), which was further increased if stratified between well-nourished and malnourished patients (*p* <0.001) suggesting that malnutrition could be a predictor that affects n-3 supplementation prolonging survival [14]. Pancreatic patients generally have a median survival of 4.1 months, yet in a study by Wigmore et al., EPA supplementation (increasing doses up to a max 6 g per day for 12 weeks) increased the median survival to 6.8 months [57]. In a metastatic breast cancer study where all patients were DHA supplemented, stratification by amount of DHA incorporated into plasma showed that higher DHA incorporation was associated with longer time to progression (8.7 months vs. 3.5 months) and overall survival increased from 18 to 34 months [7]. One g of n-3 per day for 51 days increased overall survival (30.9 ± 3.7 versus 25.9 ± 3.6 weeks, *p* = 0.05; HR = 0.41, 95% CI: 0.20–0.84) and disease-free survival (28.5 ± 3.3 versus 23.7 ± 3.6, *p* = 0.03; HR = 0.44, 95% CI: 0.22–0.87) in stage IIIB breast cancer patients [8]. Supplementation with 600 mg EPA + DHA increased time to progression in colorectal patients (20 months vs. 11 months for non-supplemented controls) [34] and overall long-term survival (at 465 days) compared to control in leukemia/lymphoma patients [48]. 

#### 3.4.6. Additional Parameters

Two of the included studies assessed the Ki67 proliferation index. Ki67 in the tumor is a clinically relevant measure of efficacy in many clinical trials as it is expressed throughout the cell cycle (G_1_, S, G_2_ and M phases, but not in G_0_) [71,72,73]. Darwito et al. found decreased Ki67 expression in breast cancer patients receiving 1 g per day of n-3 supplement (39.2 ± 5.3 versus 42.4 ± 4.8, *p* = 0.03) [8], while there were no observed changes in the Ki67 proliferation index in patients with colorectal cancer and liver metastases when supplemented with 2 g of EPA per day [36]. Other experimentally relevant markers have been assessed in supplementation studies that suggest efficacy of n-3 fatty acids in modulating cancer outcomes including decreased vascular endothelial growth factor (VEGF) [47] and oxidative status [50], increased glutathione reductase, antioxidant status [16], superoxide dismutase [16,20] and phagocytosis and H_2_O_2_ [20] in plasma or peripheral blood mononuclear cells and decreased VEGF [8], carcinoembryonic antigen [34] and gene expression of matrix metalloproteinase (MMP)-1 and MMP-9 [30] in tumors.

## 4. Discussion

The current scoping review brings together the evidence of over 50 clinical trials supporting the efficacy and antineoplastic effects of n-3 supplementation in a clinical setting. The goal of this review was to provide a broad overview of the evidence that is currently known on this topic and it is not intended to be a systematic review. Therefore, we have not provided a formal evaluation of the strength of the evidence or risk of bias for the studies included [74]. The evidence suggests that providing EPA and DHA (alone or in combination) can result in measurable clinical benefits across a spectrum of cancers. Although this research spans two decades, no clear guidelines for use as a therapeutic intervention have been defined. To establish recommendations, there is a need for future studies to focus on key factors that will strengthen the evidence currently available: (1) well-defined n-3 (EPA versus DHA) intervention, (2) clear dose based on preliminary trials, (3) expansion of outcomes assessed and (4) improved reporting of outcomes. 

### 4.1. N-3 Supplement Component and Dose Prescribed

Benefits have been reported with both EPA and DHA, alone and in combination, at a range of doses from 300 mg to 5 g per day. However, what the optimal combination of n-3 fatty acids is or what the dose should be is unclear. It is well established that EPA and DHA have both similar and unique mechanisms of action in cancer and other chronic diseases [75,76,77,78,79]. In an antineoplastic setting, while EPA more strongly inhibits arachidonic-derived prostaglandin production, DHA is known to modulate membrane lipid rafts, increase production of oxidative products and beneficially bind/activate nuclear receptors to a greater extent than EPA [76]. Currently, most studies used a combination of EPA and DHA, with EPA being the predominant fatty acid in the supplement, likely defined by the available supplements. Trial outcomes need to be clearly identified and then a single or mixed n-3 supplement used based on these parameters. 

With respect to dose, it could be that regardless of the amount prescribed, there is a maximal level of incorporation within tumoral and tissue membranes. Achieving this through a personalized medicine approach and modifiable n-3 dose could be a consideration for future studies. As plasma is more easily accessible in a study setting compared to surgical tumor samples and plasma phospholipid membrane content of n-3 fatty acids correlates with tumor content [80], it is a metric that could be followed easily in clinic. Indeed, Bougnoux et al. found that patients with higher levels of plasma DHA had a longer time to progression and increased overall survival than patients with lower plasma levels, in a study where all patients received the same dose of 1.8 g DHA per day [7]. Additionally, while many of the studies included in this review reported that compliance was evaluated, confirmation through assessment of blood components or tissue fatty acid incorporation was reported in less than 20% of the studies [7,9,12,19,20,35,36,49,50,57]. Furthermore, consistent reporting of baseline would provide valuable information on incorporation and could be important in determination of clinical guidelines. 

### 4.2. Reported Outcomes

Many outcomes have been reported in the literature, nonetheless there are three categories where future research and enhanced reporting would help strengthen the evidence currently available: (1) clinically relevant outcomes, (2) immune system modulation and (3) mechanistically relevant data.

#### 4.2.1. Clinically Relevant Outcomes

Certain clinically relevant outcomes reporting the benefits of n-3 supplementation include weight loss and cancer cachexia. However, there exists a need for high-quality evidence for other metrics that are established clinical outcomes. Currently, only a few studies have reported on serious adverse events and the impact of EPA or DHA on chemotherapy associated toxicities. Taxanes are administered for many solid carcinomas including breast, ovarian, lung and pancreatic and neuropathies occur in up to 70% of these patients [81] although it is reported to be much lower with docetaxel compared to paclitaxel [66]. There are studies that have reported a beneficial decrease in taxane related neuropathies [9,54,56]. Interestingly, two of these studies had higher doses of DHA administered (compared to the EPA dose) and it would be of interest to determine whether DHA provides protection more efficaciously. Ki67 is used as a prognostic marker in several cancer types including breast cancer [82] and adrenocortical [83] yet was only reported in two studies in this review [8,36]. This is a quantifiable metric that would enable clinicians to establish concrete guidelines if it is routinely assessed in future studies. Additionally, many studies to date have focused on stage III, IV or palliative cancer patients and provide little data on progression-free survival, overall survival, or disease-free survival. How supplementation impacts cancer outcomes for earlier stage cancers is not known and this is a key metric that should be investigated in the future. 

#### 4.2.2. Modulation of Immune Function

The immune system, inflammation and cancer are inextricably linked. Progression of several cancers including pancreatic, gall bladder and esophageal results from chronic inflammation and prevalence of tumor-associated neutrophils that are associated with poor prognosis in many cancers including colorectal, melanoma and glioblastoma [84]. An estimated 40 to 60% of lung and gastrointestinal cancer patients exhibit elevated plasma concentrations of CRP [85] and this was consistently observed in the baseline data of the studies assessed herein. To counteract the inflammatory role in development or promotion of cancers, the beneficial role of n-3 fatty acids on the immune system of a cancer patient is becoming better defined. However, a comprehensive assessment of immune response has not been reported on in the current evidence. CRP data were reported in only 30% of the studies and only one study assessed time points other than baseline and end of study [41]. Future studies would benefit from a temporal assessment of changes in CRP over the course of chemotherapy as this molecule is often use as a surrogate marker of the inflammatory response. There is currently no n-3 supplementation study that has thoroughly detailed the effects of supplementation and the complex immune response of cytokines and immune cells that occur during the cancer trajectory. Profiling of both pro-inflammatory cytokines including IL-6, IL-1β, IL-17, IL-23 and TNFα that are indicators of poor prognosis as well as anti-inflammatory cytokines including IL-4, IL-10, IL-13, and transforming growth factor (TGF)β is recommended for future studies. Finally, dendritic cell activation and natural killer cell status are becoming increasingly recognized as important host responses through the cancer trajectory and have not been explored in the context of n-3 supplementation.

#### 4.2.3. Mechanistically Relevant Outcomes

There exists a substantial base of strong preclinical evidence that details mechanisms and pathways involved in the anticancer actions of n-3 fatty acids. Mechanisms that have been identified include decreased cell proliferation, cell cycle progression and increased apoptosis and oxidative stress (reviewed in [4,5,6]). Pathways that are modulated through n-3 supplementation include the CD95 death receptor pathway, the Wnt/β-catenin pathway, the mitogen-activated protein kinase (MAPK)/ERK pathway, the phosphoinositide 3-kinase (PI3K) pathway, the Janus kinase (JAK)-Signal transducer and activator of transcription (STAT) pathway and the nuclear factor (NF)-κB pathway [4]. Moving forward, it will be important to bridge the gap between preclinical and clinical evidence. Obtaining fresh tumor samples to properly assess mechanisms involved would greatly improve the strength of the evidence available. While this is not always practical in a clinical setting, future studies could also consider immunohistochemical assessment of fixed tumor samples or analysis of circulating tumor cells in blood. 

## 5. Conclusions

The use of the n-3 fatty acids (EPA and/or DHA) as a therapeutic intervention in a clinical setting is backed by strong biological hypotheses and a large body of preclinical data. The current clinical evidence suggests that it could improve a variety of outcomes important to the patient and the disease process. There is, however, a need for larger, well-controlled, statistically powered studies with expanded reported outcomes. Furthermore, the amount and length of dose prescribed should be large enough and long enough to test the desired outcomes. Future studies should also include cancers where there are some promising preclinical data suggesting the efficacy of n-3 supplementation, such as ovarian, but for which there have been no clinical studies to date. 

## Figures and Tables

**Figure 1 cancers-13-01206-f001:**
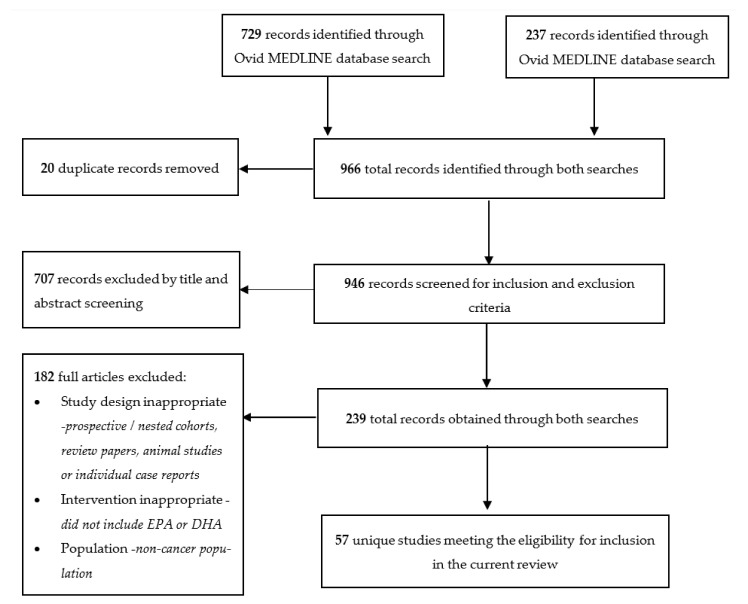
Flow chart of search and screening results for review inclusion.

**Figure 2 cancers-13-01206-f002:**
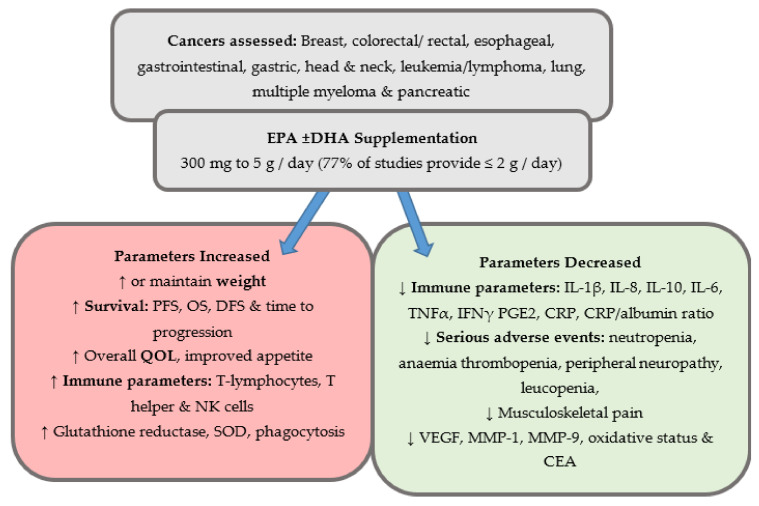
Overview of outcomes reported on the effects of n-3 supplementation compared to control or baseline status in clinical trials. Abbreviations used: CEA, carcinoembryonic antigen; CRP, C-reactive protein; DHA, docosahexaenoic acid, DFS, disease-free survival, EPA, eicosapentaenoic acid, IFN, interferon; IL, interleukin; MMP, matrix metalloproteinase, OS, overall survival, PGE2, prostaglandin E2; PFS, progression-free survival; QOL, quality of life; SOD, super oxide dismutase; TNF, tumor necrosis factor; VEGF, vascular endothelial growth factor.

**Table 1 cancers-13-01206-t001:** Randomized controlled trials providing N-3 capsule supplementation concomitant with chemotherapy.

Cancer Type (Stage)	N (Int/CNT) Female/Male	Age, Body Weight and BMI (Int/CNT)	Chemotherapy	N-3 (g Total/Day = EPA/DHA) CNT	Treatment Duration	Experimental Findings	Ref
Breast (metastatic)	25 F	Age = 58 (32–71)	Cyclophosphamide, Fluorouracil, Epirubicin	1.8 g DHA CNT = N/A	18 weeks	Stratified by amount of DHA incorporated into plasma ↑ DHA group associated with longer time to progression (8.7 months vs. 3.5 months); ↑ OS (34 vs. 18 months; ↓ neutropenia, anaemia and thrombopenia	2009 [7]
Breast	57 F (30/27)	Age = 46.2 ± 9.8/45.7 ± 12.0 BMI = 46.0 ± 9.0/44.1 ± 8.9	Paclitaxel	1.2 g = 0.19 g EPA/1.0 g DHA CNT = sunflower oil	4 cycles + 1 month post	N-3: 70% ↓ risk of peripheral neuropathy incidence	2012 [9]
Breast (I–III)	209 F (102/107)	Age = 59.5/59.1 Body weight = 79.0 (77.3–79.8)	Anastrozole, Exemestane or Letrozole	3.3 g = 2.24 g EPA/1.12 g DHA CNT = soybean and corn oil	24 weeks	Both groups: ↓ in pain symptoms in but no proof of n-3 efficacy; when stratified by BMI, n-3 significantly ↓ pain in obese patients	2015 [10,11]
Breast (IIIB)	48 F (24/24)	Age = 46.5 ± 8.1/48.5 ± 8.8	Cyclophosphamide, Doxorubicin, Fluorouracil	1.0 g N-3 CNT = unknown source	51 days	N-3: ↓ Ki67 (39.2 ± 5.3 vs. 42.4 ± 4.8, P = 0.03), ↓ VEGF (29.5 ± 5.4 vs. 32.7 ± 5.2, P = 0.04). ↑ OS (30.9 ± 3.7 vs. 25.9 ± 3.6 weeks, P = 0.05; HR = 0.41, 95% CI: 0.20–0.84 and ↑ DFS (28.5 ± 3.3 vs. 23.7 ± 3.6, P = 0.03; HR = 0.44, 95% CI: 0.22–0.87	2019 [8]
Breast (I–II)	5 F	Age = 50 (34–60)	Cyclophosphamide + Fluorouracil + Doxorubicin/Adriamycin or Paclitaxel	1.2 −1.8 g = 0.72 g–1.1 g EPA/0.48–0.72 g DHA	130–188 days	N-3: ↑ SOD, glutathione reductase and plasma antioxidant status; ↑ QOL	2015 [16]
Gastric (I–IV)	34 (17/17) 15 F/19 M	Age = 71.2 ± 9.8/67.5 ± 11.2	Cisplatin	1.25 g = 0.92 g EPA/0.32 g DHA CNT = Placebo	9 weeks	N-3: ↓ gene expression of MMP-1 and MMP-9 compared to control	2019 [30]
Gastrointestinal	38 (19/19) 16 F/22 M	Age = 53.8 ± 2.4/54.9 ± 3.2 Body weight = 65.8 ± 3.6/69.5 ± 3.6	Fluorouracil and Leucovorin	0.70 g = 0.30 g EPA/0.40 g DHA CNT = N/A	8 weeks	N-3: ↑ in EPA and DHA in PBMCs, ↑ in phagocytosis, superoxide anion production and H_2_O_2_ productions, ↑ weight, improved neutrophil function during chemotherapy Control: ↓ weight	2011 [20]
Gastrointestinal	51 (26/25) 24 F/27 M	Age = 58 (46–63)/51 (41–60) BMI = 26.5 ± 4.6/25.6 ± 4.2 weight loss = 7.2–11.3% in 6 months prior to study entry	Capecitabine + Oxaliplatin; Fluorouracil + Oxaliplatin; Fluorouracil + Leucovorin; other	1.55 g = 1.0 g EPA/0.55 g DHA CNT = olive oil 2 g	9 weeks	N-3: ↓ in severe diarrhea compared to control and better performance status score	2019 [26]
Gastrointestinal (II/III)	81 35 F/46 M	Age = 56.8 ± 10.6/59.9 ± 8.8 Body weight = 68.4 ± 9.8/68.8 ± 12.0 BMI = 24.3 ± 2.9/25.4 ± 3.6	Not stated	0.61 g = 0.11 g EPA/0.50 g DHA	8 weeks	N-3: Maintained weight compared to control; ↓ CRP compared to baseline; NS decrease in TNFα and IL-6 compared to baseline; in combination with vitamin D ↓ CRP, TNFα and IL-6 compared to baseline	2019 [27]
Colorectal	140	Body weight = 54.2 ± 11.7/57.4 ± 10.9 BMI = 21.8 ± 4.1/23.0 ± 4.3	Capecitabine + Oxaliplatin	1.40 g EPA + DHA	8 weeks	N-3: ↑ global health status, ↓ fatigue, nausea, pain, ↓ IL-6 compared to baseline and control; NC TNFα or CRP	2018 [45]
Colorectal (III and IV)	11(6/5) 5 F/6 M	Age = 53.6 ± 12.9/55.2 ± 7.7 Body weight = 72.3 ± 12.3/68.1 ± 12.1 BMI = 28.6 ± 6.3/26.4 ± 3.7	Xeloda, Oxaliplatin, Fluorouracil and/or Leucovorin	0.60 g = 0.36 g EPA/0.24 g DHA CNT = N/A	9 weeks	N-3: Improved CRP, CRP/albumin and potentially prevented weight loss	2013 [32]
Colorectal and Rectal	23(11/12) 6 F/17 M	Age = 50.1 ± 8.2/54.3 ± 9.3 Body weight = 73 ± 16.8/66.8 ± 11.6 BMI = 27.3 ± 6.1/25.0 ± 3.4	Type not specified	0.60 g = 0.36 g EPA/0.24 g DHA CNT = N/A	9 weeks	N-3: ↓ CRP/albumin ratio	2012 [33]
Colorectal (II–IV)	30 (17/13) 10 F/20 M	Age = 52.1 ± 7.6/53.1 ± 10.2	Chemotherapy type not stated	0.60 g = 0.36 g EPA/0.24 g DHA CNT = N/A	9 weeks	N-3: ↑ time to progression (20 vs. 11 months); ↓ carcinoembryonic antigen	2016 [34]
Leukemia Lymphoma	22 (9/13) 10 F/12 M	Age = 43.8/53.8 Body weight = 68.1 ± 10.3/72.4 ± 11.6 BMI = 24.6 ± 4.1/25.7 ± 4.0	Type not specified	0.61 g = 0.37 g EPA/0.24 g DHA CNT = N/A	9 weeks	N-3: ↓ CRP/albumin ratio from high to low; ↑ overall long-term survival (at 465 days) compared to control Control: ↓CRP/albumin ratio from high to medium	2017 [48]
NSCLC (III or IV)	46 (31/15) 22 F/24 M	Age = 64 ± 1.7/63 ± 2.1	Carboplatin and Vinorelbine or Carboplatin and Gemcitabine	2.4–2.7 g = 2.2 g EPA + 0.24–0.50 g DHA CNT = SOC	6 weeks	N-3: ↑ chemotherapy response rate, ↑ clinical benefit; ↑ 1-year survival (trend)	2011 [49]
Lung (Advanced)	27 (13/14) 8 F/19 M	Age = 55.6 ± 7.4/60.6 ± 7.4 Body weight = 75.1 ± 16.1/68.0 ± 12.8 BMI = 26.2 ± 7.0/25.2 ± 3.9	Gemcitabine, Cisplatin	3.4 g = 2.0 g EPA/1.4 g DHA CNT = olive oil	66 days	N-3: ↑ in EPA + DHA in plasma, ↑ in EPA in RBC; ↓ IL-6, PGE2 and ↑ Body weight; ↓ inflammatory indexes and oxidative status; Control: ↑ CRP, IL-6, TNF and ROS	2012 [50]
NSCLC (Advanced)	137 (77/60) 61 F/76 M	Age = 63.8 ± 6.4/62.9 ± 7.1 Body weight = 67.2 ± 11.5/70.1 ± 12.3 BMI = 23.5 ± 2.1/23.9 ± 2.4	Cisplatin, ±TXT, ± Bevacizumab	0.71 g = 0.5 g EPA/0.20 g DHA CNT = N/A	6 weeks	N-3 group ↓ CRP, IL-6 and PGE2; NC in QOL or nutritional status	2018 [51]
Multiple Myeloma	18 8 F/11 M	Age = 69 (57–76)	Bortezomib + Thalidomide + Dexamethasone (84 days) or Bortezomib + Melphalan + Prednisone	2 g = 1.2 g ALA + 0.80 g DHA CNT = N/A	6 months	N-3: ↓ in onset or worsening of neuropathic pain, ↓ in chemotherapy interruptions	2018 [56]

Abbreviations used: ALA, alpha linolenic acid; BMI, body mass index; CI, confidence interval; CNT, control; CRP, C-reactive protein; DFS, disease-free survival; DHA, docosahexaenoic acid; EPA, eicosapentaenoic acid; F, female; g, gram(s); H_2_O_2_, hydrogen peroxide; HR, hazards ratio; IL, interleukin; Int, intervention group; M, male; MMP, matrix metalloproteinase; N, number; N/A, not applicable; NC, no change, NS, non-significant; NSCLC, non-small-cell lung cancer; OS, overall survival; PBMC, peripheral blood mononuclear cell, PGE2, prostaglandin E2; QOL, quality of life; RBC, red blood cell; Ref, reference; SOC, standard of care; SOD, super oxide dismutase; TNFα, tumor necrosis factor; VEGF, vascular endothelial growth factor.

**Table 2 cancers-13-01206-t002:** Randomized controlled trials providing N-3 supplementation without chemotherapy.

Cancer Type (Stage)	N (Int/CNT) Female/Male	Age, Body Weight and BMI (Int/CNT)	Chemotherapy	N-3 (g EPA + DHA/Day) CNT	Treatment Duration	Experimental Findings	Ref
Breast (I–III)	37 F (18/19)	Age = 48.6 + 9.0/53.4 + 7.5 BMI = 43% overweight, 30% obese	No chemotherapy-N-3 supplementation prior to treatment	0.94 g EPA + 0.78 g DHA CNT = 2 g mineral oil	30 days	N-3: NC CD4+, CD8+, PGE2, IL-6 Control: ↓ CD4+, NC PGE2, IL-6, ↑ hsCRP	2017 [12]
Breast (I–III)	45	Age = 57.3 (40–81) BMI = 28.9 (19.3–38.3)	Previous chemotherapy (69.9%), previous radiotherapy (87%); currently on aromatase inhibitors 67.3% or Tamoxifen 32.6%	1.38 g N-3 CNT = N/A	30 days	N-3: ↓ from baseline at day 30 and day 60 of CRP; 21.5% decrease in pain scale; ↓ in IFNγ at day 30	2019 [13]
Breast, gastrointestinal, lung, liver, pancreas (all metastasized)	64 (60 completed) 24 F/36 M	Age = 60 ± 5 (F), 57 ± 4 (M)/58 ± 4 (F), 56 ± 3 (M)	Previous surgery n = 38, Previous chemotherapy n = 26, previous radiotherapy n = 6, none = 10	3.1 g EPA + 2.1 g DHA CNT = sugar tablets	until death	Both groups: ↑ in survival in well-nourished vs. malnourished N-3: ↑ in survival, ↑ CD4/CD8	1998 [14]
Colorectal (local and advanced)	30	Age = 63 ± 2.3	Surgery	Group 1 (localized): 1.2 g GLA + 1.1 g EPA + 0.16 g DHA Group 2 (advanced): T0–15 1.2 g GLA + 1.1 g EPA + 0.16 g DHA, T16–30: 1.8 g GLA + 1.6 g EPA + 0.24 g DHA; Months 2–6: 2.3 g GLA + 2.1 g EPA + 0.32 g DHA Group 3: CNT (6 months)	Group 1: until surgery Group 2 and 3: 6 months	Group 1: NC in immune parameters; Group 2 = ↓ IL1β 3, 4, 5 and 6 months; ↓ IL-4 at 2, 3, 4, 5 and 6; ↓ IL-6 at 6 months; ↓ TNFα at 2, 4, 5 and 6; ↓ IFNγ by month 4 Group 3: NC	1994 [35]
Colorectal (Dukes A–D) All with liver metastases	88 43 (17 F/26 M)/45 (10 F/35 M)	Age = 71 (35–87)/68 (44–82)	Previous chemotherapy	2 g EPA CNT = 2 g MCT	12–65 days	N-3: ↑ EPA in tumor tissue, NC in Ki67, ↑ OS at 18 months (trend)	2014 [36]
Advanced lung (III–IV)	22 (10/12) 5 F/17 M	Age = 64 (44–90)/61 (44–83) Body weight = 60.1 ± 8.2/62.8 ± 9.7 BMI = 24 ± 6.2/25.8 ± 4.4	N/A	0.36 g EPA + 0.24 g DHA + celecoxib CNT = 0.36 g EPA + 0.24 g DHA	6 weeks	N-3 + celecoxib: ↓ CRP; ↑ body weight and hand grip scores improved N-3 alone: ↓ CRP	2007 [52]
Pancreatic (II–IV)	26 14 F/12 M	Age = 56 (39–75) Body weight = 66.8 (56.0–75.1) BMI = 23.2 (21.1–27.4)	N/A	EPA only week 1: 1 g week 2: 2 g week 3: 4 g week 4–12: 6 g CNT = N/A	12 weeks until death	Body weight stabilized and began to increase by week 4; CRP stabilized or was slightly reduced in patients who had ↑ CRP at beginning; median survival = 6.8 months	2000 [57]
Pancreatic	33 (18/15) 17 F/16 M	Age = 70.3 ± 8.2/71.3 ± 7.5 Body weight = 62.9 ± 6.5/71.4 ± 15.3 BMI = 21.3 ± 1.7/23.7 ± 4.1	24 patients received chemotherapy, 2 received radiotherapy (not all curative, most palliative)	Group 1: 0.10 g EPA + 0.20 g DHA; Group 2: 0.13 g EPA + 0.18 g DHA CNT = N/A	6 weeks	↑ in HDL in Group 1	2017 [58]

Abbreviations used: BMI, body mass index; CD, cluster of differentiation; CNT, control; CRP, C-reactive protein; DHA, docosahexaenoic acid; EPA, eicosapentaenoic acid; F, female; g, gram(s); GLA, gamma linolenic acid; HDL, high-density lipoprotein; hsCRP, high sensitivity CRP; IFNγ, interferon gamma; IL, interleukin; Int, intervention group; M, male; MCT, medium chain triglycerides; N, number; N/A, not applicable; NC, no change, OS, overall survival; PGE2, prostaglandin E2; Ref, reference; SOC, standard of care; TNFα, tumor necrosis factor.

**Table 3 cancers-13-01206-t003:** Randomized controlled trials providing oral N-3 supplementation.

Cancer Type (Stage)	N (Int/CNT) Female/Male	Age Body Weight and BMI (Int/CNT)	Chemotherapy	N-3 (g EPA + DHA/Day) CNT	Treatment Duration	Experimental Findings	Ref
Lung, Head and Neck, Gynecologic, Breast, Prostate, Urinary Tract, Esophagus (I–IV)	38 20/18 14 F/24 M	Age = 62.7 ± 11.0 Body weight = 70.8 ± 12.6 BMI = 24.8 ± 3.5	Radiotherapy	2 × 326 kcal: 2.4 g EPA + 1.2 g DHA + 40 g protein CNT = N/A	7 days	N-3: ↓ serum PGE2 Control: ↑ serum PGE2 No differences in cytokine production	2013 [15]
Stomach, Colon, Lung, Pancreas, Other	40	Age = 61.3 ± 12.1/63.6 ± 11.4 BMI = 20.9 ± 3.7/22.2 ± 3.8	Chemotherapy ± radiation or no treatment	600 kcal: 1.5 g EPA + 19.5% protein CNT = isocaloric supplement	1 month	Both groups: ↑ SF36 N-3: ↓ in IFNγ Control: ↑ in IFNγ	2011 [21]
Head and Neck (I–IV)	27 (13/14) 11 F/16 M	Age = 61.5(45–77)/66.1 (47–76) BMI = NS but cachexic	Surgery	600 kcal: 2.1 g EPA + 32 g protein CNT = N/A	4 weeks	No differences between groups or from baseline	2018 [17]
Head and Neck (I–IV)	64 29 F/35 M	Age = 60 ± 14/58 ± 14 Body weight = 58.8 ± 1.4/61.1 ± 11.5 BMI = 22.6 ± 4.6/24 ± 4.2 weight loss = ~9 kg in 3 months before entry	Surgery, radiotherapy, chemotherapy, or combination	600 kcal: 2 g EPA + 40 g protein CNT = isocaloric supplement	6 weeks	N-3: weight maintenance, ↓CRP, TNFα and IFNγ Control: weight loss (2.0 ± 3.7 lbs), ↓ CRP, ↑ TNFα and IFNγ	2018 [18]
Colorectal (IV)	23 8 F/13 M	Age = 61 ± 11.6 Body weight = 75.9 ± 17.0 BMI = 28 ± 6.4	Chemotherapy 17 with previous chemotherapy	600 kcal: 2.2 g EPA + 0.92 g DHA + 32 g protein CNT = N/A	9 weeks	N-3: ↓ in GM-CSF, ↑ RANTES, CRP (week 3) ↑ in GM-CSF and NC CRP (compared to baseline; week 9), Correlations between baseline IL-10 and survival, IL-6 and survival, IL-6 and CRP	2007 [41]
Colorectal (IV)	13 (5/6) 4 F/9 M	Age = 61.5 ± 15.8/68.2 ± 15.6 Body weight = 69.9 ± 15.9/72.2 ± 11.7 BMI = 25.8 ± 4.3/26 ± 3.3	Fluorouracil + oxaliplatin + folinic acid or capecitabine	600 kcal: 2 g EPA + 0.9 g DHA + 32 g protein CNT = N/A	12 weeks	N-3: ↑ weight, NS improvement in QOL and appetite, NS ↓ in fatigue and pain	2010 [37]
Gastric (I–IV)	68 24 F/44 M	Age = 58 Body weight= 63.5 (58.1–69.8)/66.1 (71.7–75.4) BMI = 24.2 (20.4–26.3)/22.8 (20.1–28.3)	Not stated	600 kcal: 2 g EPA + 1.2 g DHA + 24 g protein CNT = isocaloric supplement	30 days	N-3: ↑ weight and ↓ IL-6 compared to control	2018 [31]
Gastrointestinal	42 15 F/27 M	Age = 68.1/66.7 Body weight = 69.1/67.8	Surgery	10.5% n-3 of 25% fat + 5.6 g protein in 100 mL (patients received 25 kcal/kg body weight)	16 days postoperative	N-3: NC in albumin, transferrin, prealbumin, PHA; ↑ stimulated IFN, CD3+, CD3 + HLA-DR, CD4+ and B lymphocytesBoth groups: ↓ T lymphocytes (preoperative to postoperative)	1995 [38]
Gastrointestinal (Advanced)	24 10 F/14 M	Age = 66 ± 9/69 ± 10 Body weight = 56.6(35–101)/61.8(33–80) BMI = 21.6 ± 4.1/21.1 ± 4.8 All had >10% weight loss in past 6 months	Palliative (at least 2 rounds of chemotherapy before study entry)	4.9 g EPA and 3.2 g DHA± melatonin CNT = isocaloric supplement	4 weeks	N-3: 38% had weight maintenance, No statistically significant changes in cytokines	2005 [39]
Gastrointestinal (II–IV)	128 38 F/90 M	Age = 72.3 ± 8.4/68.9 ± 10.3 Body weight = NS but 5% weight loss before entry	44 adjuvant chemotherapy/84 palliative chemotherapy	600 kcal supplement: 2.2 g EPA + 0.92 g DHA + 32 g protein CNT = N/A	6 months	N-3: stable CRP Control: ↑ CRP	2017 [64]
Lung, Gastrointestinal (I–IV)	69 28 F/21 M	Age = 63.5 ± 11.8 BMI = not stated but 87% moderate or severe malnutrition	Chemotherapy	600 kcal: 2.2 g EPA 33 g protein CNT = isocaloric supplement	4 weeks	N-3: ↓ CRP (NS due to dropouts/death only 18 in N-3 vs. 25 in control for final analysis)	2014 [40]
NSCLC (III)	40 19 F/21 M	Age = 58.4 ± 12.0/57.2 ± 8.1 Body weight = 77.1 ± 14.6/64.7 ± 7.4 BMI = 24.8 ± 4.1/23.0 ± 2.4	Cisplatin ± docetaxel or± bevacizumab + concurrent radiotherapy	600 kcal: 2.2 g EPA + 1 g DHA + 32 g protein CNT = isocaloric supplement	6 weeks	N-3: weight maintenance, NC in CRP, IL-6, TNFp55, albumin and HLA-DR	2012 [53]
NSCLC (III–IV)	84 (44/40) 49 F/43 M	Age = 58.8 ± 14/61.1 ± 12.4 Body weight = 60.4 ± 11/64.7 ± 11; BMI = 24.2 ± 3/25.2 ± 4 weight loss before entry = 8.8 ± 8%/7.4 ± 9%	Paclitaxel and cisplatinum	300 kcal: 1.1 g EPA + protein CNT = isocaloric supplement	6 weeks supplement and up to 18 weeks chemotherapy)	N-3: weight maintenance; ↓ CRP, TNFα; ↑ protein intake improved global health status (including fatigue and improved appetite); trend towards progression-free survival Control: weight loss, ↑ neuropathy	2014 [54]
Pancreatic, NSCLC	7 2 F/5 M	Age = 55.1 ± 5.0 = Body weight = 77.5 ± 11.5 (12% weight loss in previous 6 months)BMI = 26.8 ± 5.7	Gemcitabine ± other	300 kcal: 1.1 g EPA + 16 g protein CNT = N/A	8 weeks	N-3: ↑ in protein intake, total energy intake, body weight and QOL	2004 [55]
Pancreatic (II–IV)	36 (18/18) (+ 6 no cancer controls)	Age = 64(56–66)/60(54–70) Body weight = 55.0(46.5–60.5)/58.5(47.8–70.7); pre-study weight loss = 17.9% (15.9–20.7)/11.8% (5.6–23.5)	Palliative	2 × 610 kcal: 2.2 g EPA + 0.96 g DHA + 32 g protein CNT = N/A	24 days	Baseline: Cancer patients: ↓ albumin, prealbumin and transferrin; ↑ CRP, fibrinogen, haptoglobin, ceruloplasmin. After intervention: N-3: ↑ albumin, prealbumin, transferrin; ↓ CRP; 1.0 kg weight gain	1999 [59]
Pancreatic (II–IV)	20 10 F/10 M	Age = 62 (51–75) Body weight = 55.2 (48.8–61.2); 17.9% (15.9–22.8) weight loss BMI = 19.8 (17.8–21.8)	Palliative surgical procedures	2 × 610 kcal: 2.2 g EPA, 0.96 g DHA + 32 g protein CNT = N/A	3–7 weeks	N-3: weight gain = 1.0 kg at 3 weeks, 2 kg at 7 weeks; ↓ IL-6 in stim PBMCs and ↓ trend IL1β (P = 0.07), NC in TNFα, CRP, unstimulated production of cytokines, or serum concentrations of IL-6, sTNF-RI, sTNF-RII, or sIL-6R and NC leptin; ↑ in fasting insulin	1999, 2001 [60,61]

Abbreviations used: BMI, body mass index; CD, cluster of differentiation; CNT, control; CRP, C-reactive protein; DHA, docosahexaenoic acid; EPA, eicosapentaenoic acid; F, female; g, gram(s); GM-CSF, granulocyte macrophage colony-stimulating factor; HLADR, Human Leukocyte Antigen—DR; IFNγ, interferon gamma; IL, interleukin; Int, intervention group; kcal, kilocalorie; kg, kilogram; M, male; N, number; N/A, not applicable; NC, no change, NS, non-significant; NSCLC, non-small-cell lung cancer; PBMC, peripheral blood mononuclear cell, PGE2, prostaglandin E2; PHA, phytohaemagglutinin; QOL, quality of life; RANTES, regulated on activation, normal T cell expressed and secreted (CCL5); Ref, reference; SF36, short form (36) health survey; TNFα, tumor necrosis factor.

**Table 4 cancers-13-01206-t004:** Randomized controlled trials providing N-3 enteral or parenteral supplementation.

Cancer Type (Stage)	N (Int/CNT) Female/Male	Age Body Weight and BMI (Int/CNT)	Chemotherapy	N-3 (EPA + DHA/Day) CNT	Treatment Duration	Experimental Findings	Ref
Esophageal (O–III)	27 4 F/23 M	Age = 67 ± 3/64 ± 2	N/A	150 mg n-3/100 mL (up to max 1.5 L/day = 2.25 g) + protein CNT = EN	Day 0 and 8	N-3: NC IL-6 between grps, ↓ in IL-8 (day 1 and 3) and PGF1a (day 5)	2005 [65]
Esophageal (O–III)	53 (28/25) 5 F/28 M	Age = 62 ± 11/65.7 ± 9 Body weight = 73.6 ± 14,/77.2 ± 13 BMI = 24.6 ± 3.4/27.1 ± 4.1	Combined radiation + chemotherapy: fluorouracil and cisplatin + surgery or surgery alone	Preoperative: 2.2 g EPA enteral feed; Postoperative: 0.45 g EPA + 0.19 g DHA/100 mL ~2.25 g EPA/day and 0.95 g DHA/day oral CNT = EN	5 + 21 days	Both groups: ↑ CRP, IL-6 after surgery and ↓ after 21 d N-3: ↓ IL-10, IL-8, maintenance of lean body mass compared to control	2009 [46]
Esophageal (Palliative)	58 16 F/42 M	Age = 67 (47–80)/66 (36–81) weight = 76.5 (49–111)/70.6 (43–106)	Capecitabine + oxaliplatin + epirubicin	N-3: 0.086 g/kg 0.04 g EPA/kg/0.046 g DHA/kg	18 weeks	N-3: ↑ in partial response; ↓ in VEGF, TNFα and IL-2 (immediately following infusions); ↓ in nausea, thromboembolism, leucopenia, neutropenia,	2019 [47]
Head and neck and Esophageal (II–IV)	28 (15/13) 5 F/23 M	Age = 57.7 ± 9.9/3.3 ± 10.4 Body weight = 60.5 ± 11.6/62.5 ± 12.6 BMI = 22.0 ± 3.6/22.3 ± 4.6 All had ~10% weight loss before the study	Combined radiation + chemotherapy: fluorouracil and cisplatin	3.4 g/L EPA + DHA CNT = EN	Chemotherapy: 5–7 weeks Int: 5 days before end of chemotherapy	N-3: ↑ in CD62 L, CD15 and NK cytotoxicity ↓ in CD4, CD8, CD45RA, CD19+, TCR 𝛼/𝛽, TCR𝛾/𝛿, NK cells ↑ in PHA stimulated TNFα and PGE2 Control: similar to N-3, Genes for immune receptors, cytokines, inflammation markers and transcription factorss were differentially expressed in n-3 vs. control	2015 [19]
Gastric	40 (20/20) 12 F/28 M	Age = 59.0 ± 12.6	Surgery	Exact n-3 formulation not given + 24% protein CNT = EN	9 days	N-3: ↑ prealbumin, transferrin, IgA, IgG, IgM, CD4, CD4/CD8 ratio and IL-2; ↓ IL-6 and TNF	2005 [28]
Gastric (I–II)	46 (26/20) 20 F/26 M	Age = 59 (36–74)/50.5 (29–75) Body weight = 65(45–89)/62 (42–88) BMI = 22.5(17.8–29.7)/22.2 (15.7–28.1)	Surgery	N-3: 0.2 g/kg body weight parenteral CNT = PN	6 days	Both groups: no difference in immunological parameters by flow, VEGF or IGF1, ↑ in CRP and IL1β N-3: ↓ in total protein, albumin, prealbumin, total cholesterol postoperative; Control: ↑ in IL-6 and TNF 𝛼	2014 [29]
Colorectal	200: 4 groups n = 50 control no supplement, control+ supplement, N-3 before and after surgery and N-3 preoperative only 82 F/118 M	Age = 62.2 ± 10.4/61.8 ± 9.9/60.5 ± 11.5/63.0 ± 8.1 20 Patients with weight loss >10%	Surgery	3.3 g N-3/L (patients received 25 kcal/kg body weight) + protein CNT = EN	7 + 7 days (pre + post)	N-3: ↑ phagocytic ability of PMN compared to controls (did not drop postoperative), ↑ IL-6 postoperative, but lower compared to control; ↓ Delayed hypersensitivity and ↓ infection in supplemented groups; NC in IGs	2002 [42]
Colorectal and Rectal	42 16 F/25 M	Age = 55.8 ± 10.1/59.2 ± 10.6 Body weight = 63.5 ± 8.9,/65.4 ± 9.2 BMI = 23.4 ± 2.4/23.9 ± 2.8	Surgery	N-3: 0.2 g/kg body weight parenteral CNT = EN	7 days	Both groups ↑ IL-6 on day of surgery N-3: ↑ CD4+ and ↓ IL-6 by day 8; NS ↓ TNF	2008 [43]
Colorectal and Rectal (Duke B–C)	57 24 F/33 M	Age = 69.8 ± 10.5/70.8 ± 6.4 BMI = 22.9 ± 3.1/23.2 ± 3.6	Surgery	N-3: 0.2 g/kg body weight parenteral CNT = EN	7 days	Both groups: ↓ CD4 on day 8 vs. day 1 N-3: ↓ CD8 day 1 and day 8; ↓ IL6 at day 8 compared to control	2012 [44]
Gastrointestinal	18 7 F/11 M	Age = 69.8 ± 2.7/65.4 ± 4.2 Body weight = 67.5 ± 4.5/59.6 ± 3.0 25% had moderate to severe protein calorie malnutrition	Surgery	N-3: 3.98 g = 2.74 g EPA, 1.24 g DHA CNT = EN	7 days	N-3: ↓ in ALT, AST and Alkaline phosphatase, ↓ in PGE2 production in LPS stimulated cells	1997 [22]
Gastrointestinal	50 20 F/30 M	Age = 62.5 ± 11.3/60.9 ± 12.5 11 patients with weight loss >10%	Surgery	N-3: 10.5% of 28% fat in 100 mL (patients received 25 kcal/kg body weight) + protein CNT = EN	7 + 7 days (pre + post)	N-3: ↑ prealbumin and retinol binding protein and ↓ IL-6, IL-1RII and delayed hypersensitivity at day 8, NC in IGs.	1999 [23]
Gastrointestinal	48 17 F/31 M	Age = 55.2 ± 12.1/52.6 ± 9.8	Surgery	146 kj/kg/day: 100 mL = 125 kcal = 0.08 g EPA + 0.03 g DHA + 4 g protein CNT = EN	7 + 7 days (pre + post)	Both groups: ↑ PGE2 and CRP postoperatively N-3: ↓ PGE2, CRP IL-6 and TNF by day 8, NS ↓ in IL2, ↑ glutamine and arginine; ↓ in CD3+, CD4+ CD8+ and NK cells at day 1 and ↑ compared to baseline and compared to control at day 8	2001 [24]
Gastrointestinal (II–III)	204 73 F/131 M	Age = 56.3 ± 10.1/58.2 ± 11.0 Body weight = 64.2 ± 10.1/64.7 ± 10.0 BMI = 22.8 ± 2.6/23.1 ± 3.1	Surgery	N-3: 0.2 g/kg body weight parenteral CNT = PN	8 days	N-3: ↓ in CD8 and NS ↓ in IL-6 and TNF compared to control at day 8	2010 [25]
Pancreatic (Advanced)	50 (20 F/30 M)	Age = 68 (40–83)	Gemcitabine	N-3: up to 500 mL (4.3–8.6 g of EPA + DHA) 1/week CNT = N/A	Up to 6 cycles (24 weeks)	N-3: ↑ in perceived QOL; 10% ↑ in global health in 47% of patients	2015 [62]
Pancreatic/Bile Duct	27 (11 F/16 M)	Age = 68.8 ± 4.24	FOLFIRINOX; gemcitabine + nabpaclitaxel or gemcitabine + TS1; TSI alone; gemcitabine alone or cisplatin + irinotecan	N-3: 2–4 packs (200 kcal/300 mg N-3/pack) = 0.60–1.20 g N3/day CNT = N/A	8 weeks	N-3: ↑ in skeletal muscle mass compared to baseline; ↑ in NK cells at week 8; trend towards increase body weight	2018 [63]

Abbreviations used: ALT, alanine aminotransferase; AST, aspartate transaminase; BMI, body mass index; CD, cluster of differentiation; CNT, control; CRP, C-reactive protein; DHA, docosahexaenoic acid; EPA, eicosapentaenoic acid; EN, standard enteral nutrition; F, female; g, gram(s); IG, immunoglobulin; IGF1, insulin-like growth factor 1; IL, interleukin; Int, intervention; kcal, kilocalorie; kg, kilogram; LPS, lipopolysaccharide; M, male; ml, millilitre; N, number; N/A, not applicable; NC, no change, NK, natural killer; NS, non-significant; PGE2, prostaglandin E2; PGF1a, prostaglandin F1a; PHA, phytohaemagglutinin; PMN, polymorphonuclear leukocytes; PN, standard parenteral nutrition; QOL, quality of life; Ref, reference; TCR, T-cell receptor; TNFα, tumor necrosis factor; VEGF, vascular endothelial growth factor.

## Data Availability

Not applicable.

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
