# Peer review of "N-3 Long-Chain Polyunsaturated Fatty Acids, Eicosapentaenoic and Docosahexaenoic Acid, and the Role of Supplementation during Cancer Treatment: A Scoping Review of Current Clinical Evidence"

_cancers, 2021, doi:10.3390/cancers13061206_

Round 1

Reviewer 1 Report

This work is an interesting scoping review very well written. Even if there is not a statistical evaluation of the strength of the included studies and no analysis of bias, it provides interesting evidences of positive role of omega-3 supplementation in different types of cancer patients.

Furthermore the discussion point out the problems related to the trials, such as dose, composition and the lack of  measures related to tissue incorporation and effects on immune and cytokines systems.

Only a minor comment is related to study selection,  I suggest to better explain the eligibility criteria that were used to exclude 707 out of 946 studies.

Author Response

Only a minor comment is related to study selection, I suggest to better explain the eligibility criteria that were used to exclude 707 out of 946 studies.

Thank you for this suggestion. We have added the following to the methods section (Lines 95-97): After evaluation of the titles and abstracts, an additional 707 articles were excluded after for failing to meet the inclusion and exclusion criteria.

The inclusion / exclusion criteria are detailed in section 2.1 therefore we have not repeated them in this section of the manuscript, however we have amended the figure to include the information.

Reviewer 2 Report

Please, unify the units – g vs. gram – for example, lines 132 – 134 (2.2 g and 0.9 g per day respectively) with a range of 2 - 3.6 grams total n-3 per day (1.1 - 2.4 grams EPA ± 0.9 - 1.2 g DHA) and in one study 4.9 grams of EPA + 3.2 grams of DHA per.

I recommend using only g for the whole text. Moreover, singular or plural forms do not need to be taken into account in the case of the abbreviation.

Similarly, in one case the authors use one way of expression of daily dose e.g. 2.2 g per day, and then another one -  2.2 g/per day. In my opinion, a slash by itself means “per”, so the right form is just 2.2 g/day (or 2.2 per day, of course).

line 133 - 2 - 3.6 grams of total n-3 per day

Figure 2 – unexplained abbreviation – QOL, which I suppose is quality of life

Tables 1 – 4: These tables might be considered as the most valuable parts of the text, bringing relevant information. For this reason, they should be thoroughly revised and unified to maximum insights. Among other things, there are some unexplained terms, abbreviations. The adding of the list of used abbreviations is also strongly needed.

For example: sunflower (oil?); CNT vs. control; weeks vs. wks; GLA; RBC; PBMCs; TCR; op; m; groups vs. grps etc. etc.

I understand the fact that many of these abbreviations are de facto notorious, but it is impossible to not explain them and even more, create new ones, or change the way of expression during the table.

Revision of used abbreviations in the whole text is completely necessary.

For instance, the first explanation of natural killer cells (NK cells) is in chapter 3.4.3., page 16, but the first use of the abbreviation is on page 5. Later, on page 20, the whole term “natural killer” is suddenly used. Other examples of the inconsistency of expression: ECOG, NSAID.

In conclusion, I would like to emphasize that although some terms or abbreviations might be considered as common and well known, particularly in a review article their adequate clarification is crucial. The careful revision of this could therefore importantly increase the value of this manuscript.

A complication of the revision was that the line numbering was missing from page 15 to the end of the text. It is therefore difficult to list where the item, which the reviewer wanted to point out, is located.

Author Response

Please, unify the units – g vs. gram – for example, lines 132 – 134 (2.2 g and 0.9 g per day respectively) with a range of 2 - 3.6 grams total n-3 per day (1.1 - 2.4 grams EPA ± 0.9 - 1.2 g DHA) and in one study 4.9 grams of EPA + 3.2 grams of DHA per.

I recommend using only g for the whole text. Moreover, singular or plural forms do not need to be taken into account in the case of the abbreviation.

Thank you for this suggestion. We have gone through the entire manuscript and removed gram/ grams and replace it with ‘g’ to be consistent throughout the manuscript.

Similarly, in one case the authors use one way of expression of daily dose e.g. 2.2 g per day, and then another one -  2.2 g/per day. In my opinion, a slash by itself means “per”, so the right form is just 2.2 g/day (or 2.2 per day, of course).

We have removed the / in this instance.

line 133 - 2 - 3.6 grams of total n-3 per day

We have added in the ‘of’

Figure 2 – unexplained abbreviation – QOL, which I suppose is quality of life

We have added in the abbreviation.

Tables 1 – 4: These tables might be considered as the most valuable parts of the text, bringing relevant information. For this reason, they should be thoroughly revised and unified to maximum insights. Among other things, there are some unexplained terms, abbreviations. The adding of the list of used abbreviations is also strongly needed.

For example: sunflower (oil?); CNT vs. control; weeks vs. wks; GLA; RBC; PBMCs; TCR; op; m; groups vs. grps etc. etc.

I understand the fact that many of these abbreviations are de facto notorious, but it is impossible to not explain them and even more, create new ones, or change the way of expression during the table.

Thank you for this comment. We apologize for this oversight and we have adjusted all tables to correct for the lack of abbreviations.

Revision of used abbreviations in the whole text is completely necessary.

For instance, the first explanation of natural killer cells (NK cells) is in chapter 3.4.3., page 16, but the first use of the abbreviation is on page 5. Later, on page 20, the whole term “natural killer” is suddenly used. Other examples of the inconsistency of expression: ECOG, NSAID.

In conclusion, I would like to emphasize that although some terms or abbreviations might be considered as common and well known, particularly in a review article their adequate clarification is crucial. The careful revision of this could therefore importantly increase the value of this manuscript.

A complication of the revision was that the line numbering was missing from page 15 to the end of the text. It is therefore difficult to list where the item, which the reviewer wanted to point out, is located.

Thank you for these suggestions. We have carefully reviewed all the text and clarified any abbreviations that were left out.

Reviewer 3 Report

In this review article, the authors summarized the clinical evidence for n-3 PUFA supplementation in cancer treatments and provided sufficient informations. I enjoyed reading the manuscript. However, I would have expected an analysis of the evidence and risk of bias (at least statistic data of the studies should be provided).

In the tables, it should indicated if n-3 PUFA are from fish oil, purified as TG or free-Fatty acids. 

Minor points:

Data of Globocan 2020 are now available and can be used instead of 2018

Figure 2 :  title for green and red inserts could be useful. QOL (quality of life) abbreviation is not listed.

Author Response

In this review article, the authors summarized the clinical evidence for n-3 PUFA supplementation in cancer treatments and provided sufficient informations. I enjoyed reading the manuscript. However, I would have expected an analysis of the evidence and risk of bias (at least statistic data of the studies should be provided).

Thank you for this suggestion. We have followed the guidelines detailed in Munn et al (Systematic review or scoping review? Guidance for authors when choosing between a systematic or scoping review approach. BMC Med. Res. Methodol. 2018, 18.) and addressed the limitation of a scoping review in the discussion line 353: “The goal of this review was to provide a broad overview of the evidence that is currently known on this topic and it is not intended to be a systematic review. Therefore, we have not provided a formal evaluation of the strength of the evidence or risk of bias for the studies included”.  

In the tables, it should indicated if n-3 PUFA are from fish oil, purified as TG or free-Fatty acids. 

This is an excellent suggestion and we believe it would add valuable information to the manuscript. As the source of n-3s in most studies was from fish oil, rather than adding this into the table, we have instead added the information directly into the body of the manuscript. Lines 120-123 read: Of the 57 independent studies assessed, two trials reported supplementation with DHA alone (in triglyceride form from an algal source [7] or titrated from fish oil [56], two studies employed EPA alone (as free fatty acids [36, 57] and 48 studies used a combination of EPA and DHA (derived from fish oil).  

Minor points:

Data of Globocan 2020 are now available and can be used instead of 2018

Thank you. We have updated the introduction with the most current statistics. Lines 43-47 read: In 2020, an estimated 19.3 million cases of cancer were diagnosed worldwide. The most frequently diagnosed cancers across both sexes was breast (11.7%), followed by lung (11.4%),  colorectal  (10%) and prostate cancer (7.3%) [1]. Despite advances in diagnosis and treatment, cancer accounted for an estimated 10 million deaths globally in 2020 [1].

Figure 2:  title for green and red inserts could be useful. QOL (quality of life) abbreviation is not listed.

We have added section headings and updated the abbreviation list.